# Transcriptomics-Based Repositioning of Natural Compound, Eudesmin, as a PRC2 Modulator

**DOI:** 10.3390/molecules26185665

**Published:** 2021-09-18

**Authors:** Sang Ah Yi, Ki Hong Nam, Min Gyu Lee, Hwamok Oh, Jae Sung Noh, Jae Kyun Jeong, Sangwoo Kwak, Ye Ji Jeon, So Hee Kwon, Jaecheol Lee, Jeung-Whan Han

**Affiliations:** 1School of Pharmacy, Sungkyunkwan University, Suwon 16419, Korea; angelna1023@hanmail.net (S.A.Y.); nam6422@hanmail.net (K.H.N.); sinnamonic@gmail.com (M.G.L.); moogkho@hanmail.net (H.O.); njs2429@naver.com (J.S.N.); theresia24@naver.com (J.K.J.); kwak920928@naver.com (S.K.); yejijeon83@gmail.com (Y.J.J.); jaecheol@skku.edu (J.L.); 2College of Pharmacy, Yonsei Institute of Pharmaceutical Sciences, International Campus, Yonsei University, Incheon 21983, Korea; soheekwon@yonsei.ac.kr; 3Department of Biopharmaceutical Convergence, Sungkyunkwan University, Suwon 16419, Korea; 4Biomedical Institute for Convergence at SKKU (BICS), Natural Sciences Campus, Sungkyunkwan University, Suwon 16419, Korea

**Keywords:** eudesmin, transcriptome, polycomb repressive complex 2, wnt, pluripotency

## Abstract

Extensive epigenetic remodeling occurs during the cell fate determination of stem cells. Previously, we discovered that eudesmin regulates lineage commitment of mesenchymal stem cells through the inhibition of signaling molecules. However, the epigenetic modulations upon eudesmin treatment in genomewide level have not been analyzed. Here, we present a transcriptome profiling data showing the enrichment in PRC2 target genes by eudesmin treatment. Furthermore, gene ontology analysis showed that PRC2 target genes downregulated by eudesmin are closely related to Wnt signaling and pluripotency. We selected *DKK1* as an eudesmin-dependent potential top hub gene in the Wnt signaling and pluripotency. Through the ChIP-qPCR and RT-qPCR, we found that eudesmin treatment increased the occupancy of PRC2 components, EZH2 and SUZ12, and H3K27me3 level on the promoter region of *DKK1*, downregulating its transcription level. According to the analysis of GEO profiles, DEGs by depletion of Oct4 showed an opposite pattern to DEGs by eudesmin treatment. Indeed, the expression of pluripotency markers, Oct4, Sox2, and Nanog, was upregulated upon eudesmin treatment. This finding demonstrates that pharmacological modulation of PRC2 dynamics by eudesmin might control Wnt signaling and maintain pluripotency of stem cells.

## 1. Introduction

The maintenance or differentiation of embryonic stem cells (ESCs) or adult stem cells requires exquisite control of gene expression patterns. Polycomb group (PcG) proteins play a critical role during early development by operating the transcriptional modulation of specialized gene sets [1]. Among the two main PcGs, polycomb repressive complex 2 (PRC2) contributes to chromatin compaction by catalyzing the methylation of histone H3 at lysine 27 (H3K27me3) [2]. ESCs deficient in PRC2 core components, including EZH2, SUZ12, or EED, fail to differentiate properly in vitro [3,4]. In addition, depletion of PRC2 components in mouse induced embryonic lethal disrupting early development [5,6]. However, how PRC2 mediates fine-tuning of multilineage differentiation has not been completely understood. Pharmacological modulation of PRC2 would provide a useful tool to expand our perspective on its function in developmental processes of mammalian ESCs.

Previously, we found that eudesmin, a novel inhibitor of mTOR/S6K1 pathway, contributes to determination of mesenchymal stem cell fate [7]. In this process, eudesmin treatment promoted transcription of *Wnt6*, *Wnt10a*, and *Wnt10b* genes, which disturb adipogenesis and induce osteogenic or myogenic commitment. While these non-canonical Wnt genes are related to lineage specification of mesenchymal cells [7], canonical Wnt/β-catenin signaling plays critical roles during embryonic development and homeostatic self-renewal of somatic cells [8]. Interestingly, crosstalk between PRC2 and Wnt signaling contributes to transcriptional regulation of genes that are involved in stemness and development into diverse lineage cells [9,10,11].

Although many studies have reported many therapeutic effects of eudesmin, such as anti-tumor [12,13], anti-fungal [14], anti-bacterial [15], and anti-inflammatory [16] effects, the genomewide analysis in which eudesmin controls multiple sets of genes has not been performed. In this study, we elucidated that eudesmin modulates the expression of PRC2 target genes through comprehensive transcriptomic analysis of our RNA-sequencing data and Enrichr (Figure 1A). Especially, the epigenetic enrichment of PRC2-mediated H3K27me3 controlled the expression of Wnt signaling-related genes and stemness-related genes by eudesmin treatment. Finally, we confirmed the epigenetic regulation of Wnt-related gene (*DKK1*) and pluripotency marker genes (*Oct4*, *Nanog* and *Sox2*) through RT-qPCR and ChIP-qPCR. Thus, we repurposed eudesmin as a novel modulator of the PRC2-Wnt cascade that can maintain ESC stemness.

## 2. Results

### 2.1. Enrichment in PRC2 Target Genes by Eudesmin Treatment

To investigate genomewide effects of eudesmin, we evaluated differentially expressed genes (DEGs) in 10T1/2 mesenchymal stem cells treated with eudesmin through RNA-sequencing analysis. A scatter plot of the RNA-sequencing showed the distribution of DEGs (fold change > 2) upon eudesmin treatment (Figure 1B), and the gene category plot showed the gene ontology enrichment for DEGs (Figure 1C). A heatmap showed the hierarchical clustering of upregulated and downregulated DEGs (Figure 2A). To identify epigenetic factors that are linked to DEGs by eudesmin, we utilized the transcription category of Enrichr, which organizes integrative enrichment signatures with already existing libraries. First, significantly upregulated genes by eudesmin were subjected to Enrichr. The analysis results from ENCODE Histone Modifications 2015 (Figure 2B) and Epigenomics Roadmap HM ChIP-seq (Figure 2C) databases showed that H3K27me3 occupies much of upregulated genes, designated as Cluster 1 (Figure 2D and Table 1). 

As in the case of genes upregulated by eudesmin, H3K27me3 occupies most of the downregulated genes according to Epigenomics Roadmap HM ChIP-seq (Figure 3A) and ENCODE Histone Modifications 2015 (Figure 3B) databases. Moreover, analyzing transcriptional machinery with ChIP-x Enrichment Analysis (ChEA) and Embryonic Stem Cell Atlas Pluripotency Evidence (ESCAPE) computationally predicted that components of PRC2, such as SUZ12, EED, and EZH2, can target genes downregulated by eudesmin (Figure 3C,D), which were designated as Cluster 2 genes (Figure 3E and Table 2). Taken together, through the comprehensive analysis of gene expression profiles in eudesmin-treated cells and publicly available data, we could extract DEGs by eudesmin treatment, which are under the control of PRC2-mediated H3K27me3.

### 2.2. PRC2 Target Genes Downregulated by Eudesmin Are Involved in Wnt Signaling

Based on the abundance of PRC2 target genes among DEGs by eudesmin, we next investigated the functional ontology by entering the list of Cluster 2 genes to Enrichr. Cross-analysis with Reactome 2016 showed that Cluster 2 genes are involved in various signaling pathways including Wnt, TGFBR1, and SMAD2/3 pathways (Figure 4A). Among them, the most prevalently presented pathway was Wnt signaling pathway, marked with red asterisks (signaling by Wnt, Mis-spliced LRP5 mutants have enhanced beta-catenin dependent signaling, TCF dependent signaling in response to WNT, and negative regulation of TCF-dependent signaling by WNT ligand antagonists). Cross-analysis with BioCarta 2016 (Figure 4B) or NCI-Nature 2016 (Figure 4C) also showed the high enrichment of Wnt signaling pathway in Cluster 2 genes. 

Through this gene ontology selection, we focused on the epigenetic regulation of *DKK1*, which was extracted from Wnt signaling-related genes of Cluster 2 genes (Figure 4A–C). Considering that Cluster 2 genes were PRC2 target genes, we performed ChIP assay with H3K27me3 and PRC2 components, EZH2 and SUZ12. Occupancies of EZH2, SUZ12, and H3K27me3 on the promoter region of *DKK1* were significantly increased by eudesmin treatment (Figure 4D). The PRC2-mediated H3K27me3 reduced both mRNA (Figure 4E) and protein (Figure 4F) expression level of DKK1. Based on our previous report demonstrating the inhibitory effects of eudesmin on S6K1 signaling [7], we detected the level of phosphorylated S6 (P-S6) to confirm whether eudesmin exhibited its pharmacological activity (Figure 4F). These data suggest that eudesmin regulates Wnt signaling-related genes including *DKK1* through PRC2-dependent silencing.

### 2.3. Eudesmin Induces the Expression of Pluripotency Marker Genes

In addition to Wnt signaling, another interesting ontology, ‘POU5F1 (OCT4), SOX2, NANOG repress genes related to differentiation’, showed high score in Cluster 2 genes (Figure 4A, green asterisk). Thus, we investigated the expression pattern of several genes of Cluster 1 and Cluster 2 in Oct4-depleted ESCs through the analysis of GEO profile (GDS1824) (Figure 5A,B). Transcription of four upregulated genes by eudesmin (*TET1*, *NCAN*, *COCH*, and *NOTCH3*) were reduced in Oct4 knockdown cells (Figure 5A), whereas four genes downregulated upon eudesmin treatment (*DKK1*, *ROR2*, *MBP*, and *HOXC8*) were upregulated (Figure 5B). Based on these opposite effects on transcriptomic patterns between eudesmin treatment and Oct4 knockdown, we hypothesized that eudesmin would affect pluripotency of ESCs. Thus, we next assessed the expression of pluripotency markers after treating H7 ESCs with eudesmin. Treatment of H7 cells with eudesmin (40 and 80 μM) decreased the level of phosphorylated S6K1 and phosphorylated S6 (Figure 5C) as we observed earlier [7]. The protein level of OCT4, a representative pluripotency marker, was elevated by eudesmin treatment in dose-dependent manner (Figure 5C). Consistently, the mRNA levels of pluripotency markers (*NANOG*, *OCT4*, and *SOX2*) were significantly increased by eudesmin (Figure 5D). These data suggest that eudesmin contributes to maintaining pluripotency, which is the most distinctive identity of ESCs.

## 3. Discussion

EZH2, a catalytic subunit of PRC2, leads to transcriptional silencing of multiplex genes by inducing H3K27me3. In many cancers, EZH2-mediated silencing of tumor suppressor genes is considered as a potential mechanism of cancer progression [17]. Moreover, hyperactivation and overexpression of EZH2 has been found in diverse malignant tumors including prostate, uterine, gastric, breast cancers, and lymphoma [18,19]. Hence, many researchers have tried to discover potent inhibitors of EZH2 which can be used as anti-cancer agents [20]. In 2020, the FDA approved the use of tazemetostat (Tazverik^TM^), a small molecule inhibiting EZH2, for the treatment of epithelioid sarcoma [21]. In addition to the synthetic compounds, numerous studies have identified naturally derived EZH2 inhibitors including curcumin, epigallocatechin-3-gallate, triptolide, and sulforaphane [22]. Here, we repurpose a natural compound, eudesmin, as a modulator of EZH2-mediated gene regulation through transcriptome-based analysis. One of the recent papers reported that eudesmin also downregulates EZH2 expression in cancer cells [23]. However, our genomewide transcriptomic analysis data indicate that eudesmin is not a simple inhibitor of EZH2, but it can finely control the expression of specific gene sets. Many of the genes downregulated upon eudesmin treatment were target genes of PRC2-mediated H3K27me3. Specifically, the expression of PRC2 target genes, which are involved in ESC maintenance and Wnt signaling, was significantly altered by eudesmin treatment. Taken together with an earlier study that observed EZH2 inhibitory effects of eudesmin [23], we concluded that the effects of eudesmin on EZH2 action seem to be different for each cell type and each gene. At least in stem cells, eudesmin treatment predominantly promoted the gene-suppressing action of EZH2 rather than inhibiting it.

Our earlier study identified the gene-regulatory role of mTOR/S6K1 in which S6K1 promotes H3K27me3 by inducing H2BS36p and subsequent EZH2 recruitment [24]. Additionally, we previously demonstrated that eudesmin inhibits the mTOR/S6K1 signaling pathway [7]. Thus, we could hypothesize that eudesmin would affect the H3K27me3-dependent gene silencing mechanism. However, unexpectedly, here we showed that eudesmin treatment inhibited S6K1 signaling but enhanced the level of H3K27me3. This finding highlights that the effects of eudesmin control the signaling-epigenome network in opposite directions. These results demonstrate that the increase in EZH2-mediated gene suppression upon eudesmin treatment was apparently independent of S6K1 inhibition. As we here did not assess accurate structure–activity relationship (SAR), we cannot assert whether the altered EZH2 target gene expression by eudesmin was mediated by direct competitive inhibition of EZH2, as in other synthetic EZH2 inhibitors, or via indirect interruption EZH2 action. Further works study SAR activity would provide a better understanding of the molecular mechanism underlying EZH2 target gene regulation upon eudesmin treatment. 

Another important issue of our current study is the effects of eudesmin on ESC maintenance. Multiple Wnt signaling-related genes were downregulated by eudesmin treatment. Our ChIP data showed that eudesmin suppressed the expression of *DKK1*, an endogenous antagonist of canonical Wnt signaling, by promoting the recruitment of PRC2 components to *DKK1* promoter region and thereby increasing H3K27me3. It has been reported that the repression of *DKK1* by Nanog activates Wnt/β-catenin pathway and this crosstalk between Nanog and Wnt is essential during somatic cell reprogramming [25]. We also observed that gene expression patterns by Oct4 depletion was opposite to eudesmin-treated cells. Finally, eudesmin treatment increased the expression of pluripotency markers, *NANOG*, *OCT4*, and *SOX2*, in human ESCs. Given that these three components are the core transcription factors of other pluripotency markers governing the pluripotency and self-renewal of ESCs [26], eudesmin treatment would be desirable to maintain distinctive characterizations of pluripotent stem cells in epigenetic levels. 

## 4. Materials and Methods 

### 4.1. Antibodies and Reagents

Anti-DKK1 (Santa Cruz Biotechnology, SC-374574), anti-S6 (Cell Signaling Technology, Danvers, MA, USA; #2217), anti-phospho (S235/236) S6 (Cell Signaling Technology, Danvers, MA, USA; #4856), Anti-p70 S6K1 (Santa Cruz Biotechnology, Dallas, TX, USA; SC-230), anti-phospho (T389) p70 S6K1 (Cell Signaling Technology, Danvers, MA, USA; #9205), anti-Oct4 (abcam, Cambridge, UK; ab19857), anti-actin (Millipore, mab1501), anti-histone H3 Lys27 trimethylation (Millipore, Burlington, MA, USA; 07-449), anti-EZH2 (Millipore, Burlington, MA, USA; 07-689), and anti-SUZ12 (abcam, Cambridge, UK; ab12073) antibodies were used in this study. Eudesmin was purchased from ChemFaces (Wuhan, Hubei, China; CFN96266).

### 4.2. Cell Lines and Culture Condition

C3H10T1/2 (10T1/2) cells (ATCC, Manassas, VA, USA; CCL-226), a mouse mesenchymal stem cell line, were grown in Dulbecco’s Modified Eagle’s Medium (DMEM) supplemented with 10% fetal bovine serum (FBS) and 1% penicillin/streptomycin (P/S). H7 cells (WiCell Research Institute, Madison, WI, USA; WA07), a human embryonic stem cell line, were grown on Matrigel-coated plates in TeSR-E8 medium (STEMCELL Technologies, Vancouver, BC, Canada). The medium was changed every day.

### 4.3. RNA Sequencing 

RNA was extracted from 10T1/2 cells treated with eudesmin (80 μM) for 24 h with Easy Blue reagent according to the manufacturer’s protocol (Intron Biotechnology, Seongnam, Korea; #17061). Library construction was performed using Quant-Seq library Prep kit (Lexogen). The quality of the libraries, including size distribution and molarity, was assessed on Agilent 2100 Bioanalyzer (Agilent, Santa Clara, CA, USA). The libraries were then multiplexed and sent for sequencing on an Illumina NextSeq 500 (Illumina, San Diego, CA, USA). Reads were aligned to the mouse reference genome (UCSC mm10). Genes with a *p*-value under 0.01 and a fold change above 2 were selected for further analysis.

### 4.4. Enrichr Analyses of Differentially Expressed Genes by Eudesmin

To assess the biological and epigenetic characterization of differentially expressed genes by eudesmin treatment, we used Enrichr (http://amp.pharm.mssm.edu/Enrichr/) [27]. Analyses of upregulated genes and downregulated genes were performed separately. 

ENCODE Histone Modifications 2015 and Epigenomics Roadmap HM ChIP-seq databases in Enrichr show histone modifications which are enriched in the submitted genes. ChEA 2016 and ESCAPE databases in Enrichr show transcriptional factors or epigenetic modulators that have occupancy sites for the submitted genes. Finally, Reactome 2016, BioCarta 2016, and NCI-Nature 2016 databases in Enrichr show signaling pathways that are involved in the submitted genes. *p*-values were indicated for each bar.

### 4.5. Reverse Transcription (RT) and Quantitative Real-Time PCR (qPCR) 

Total RNA extracted from cells using Easy-Blue reagent (Intron Biotechnology, Seongnam, Korea; #17061) was reverse transcribed into cDNA using a Maxime RT Premix (Intron Biotechnology, Seongnam, Korea; #25081). Quantitative real-time PCR was performed using KAPA SYBR^®^ FAST qPCR Master Mix (Kapa Biosystems, Wilmington, MA, USA; #KK4602) and a CFX96 Touch^TM^ or Chromo4^TM^ real-time PCR detector (Bio-Rad, Hercules, CA, USA). Relative levels of mRNA were normalized to the levels of *GAPDH* mRNA for each reaction. The RT-qPCR primer sequences used are as follows: *GAPDH* forward, 5′-GAGTCAACGGATTTGGTCGT-3′; *GAPDH* reverse, 5′-TTGATTTTGGAGGGATCTCG-3′; *DKK1* forward, 5′-TTCCATTTTTGCAGTAATTCCC-3′; *DKK1* reverse, 5′-AGTACTGCGCTAGTCCCACC-3′; *NANOG* forward, 5′-TGAACCTCAGCTACAAACAGGTG-3′; *NANOG* reverse, 5′-AACTGCATGCAGGACTGCAGAG-3′; *OCT4* forward, 5′-CTTGCTGCAGAAGTGGGTGGAGGAA-3′; *OCT4* reverse, 5′-CTGCAGTGTGGGTTTCGGGCA-3′; *SOX2* forward, 5‘-AGAACCCCAAGATGCACAAC-3′; *SOX2* reverse, 5‘-CGGGGCCGGTATTTATAATC-3′.

### 4.6. Chromatin Immunoprecipitation and Real-Time PCR (ChIP-qPCR)

Chromatin immunoprecipitation was performed as previously described [28]. Briefly, a small portion (5%) of the cross-linked, sheared chromatin solution was reserved as the input DNA, and the remainder was incubated with the appropriate primary antibodies and protein A agarose/salmon sperm DNA (Millipore, Burlington, MA, USA; #16-157) overnight at 4 °C. After immunoprecipitation, the chromatin fragments were de-crosslinked, eluted, and subjected to qPCR using primers specific for *DKK1* promoter. The primer sequences used for ChIP-qPCR are as follows: *DKK1* forward, 5′-TGGGCTGGCCATCTCCACCG-3′; *DKK1* reverse, 5′-ACTCCCCAGTACAAACAGAGGGCA-3′. Each amount of chromatin was normalized by input and the relative values of IgG-Con were presented.

### 4.7. Immunoblotting 

Cells were lysed in Pro-Prep (Intron Biotechnology, #17081) and each protein was subjected to SDS-polyacrylamide gel electrophoresis (PAGE). Separated proteins were transferred to polyvinylidene difluoride (PVDF) membranes using semi-dry transfer (Bio-Rad). The membranes were incubated overnight with the indicated primary antibodies, followed by incubation with horseradish peroxidase-conjugated secondary antibodies (Abcam, Cambridge, UK) for 1 h. The signals were detected using chemiluminescence reagents (Abclon, Guro, Korea). The signal of immunoblot band was quantified with ImageJ software (ImageJ bundled with 64-bit Java 1.8.0_172) and normalized by the signal of actin.

### 4.8. Public Data Availability 

The accession number for the deposited data we used in this study is GDS1824.

### 4.9. Statistical Analysis 

Statistical significance was determined through Student’s *t*-test (two-tailed) and assessed based on the resulting *p*-value. Data values are presented as the mean ± standard error of the mean (SEM) for n = 3. * *p* < 0.05, ** *p* < 0.01, *** *p* < 0.001.

## 5. Conclusions

Although the physiological function of PRC2 is well established both in development and disease progression, the use of PRC2 inhibiting compounds has been focused on cancer therapy. In this current study, we show the modulating action of eudesmin on the gene transcription under the control of PRC2. Considering that genes related to pluripotency were highly enriched on DEGs by eudesmin treatment, eudesmin can be repositioned as an epigenetic modulating agent for stem cell maintenance or reprogramming of somatic cells into induced pluripotent stem cells.

## Figures and Tables

**Figure 1 molecules-26-05665-f001:**
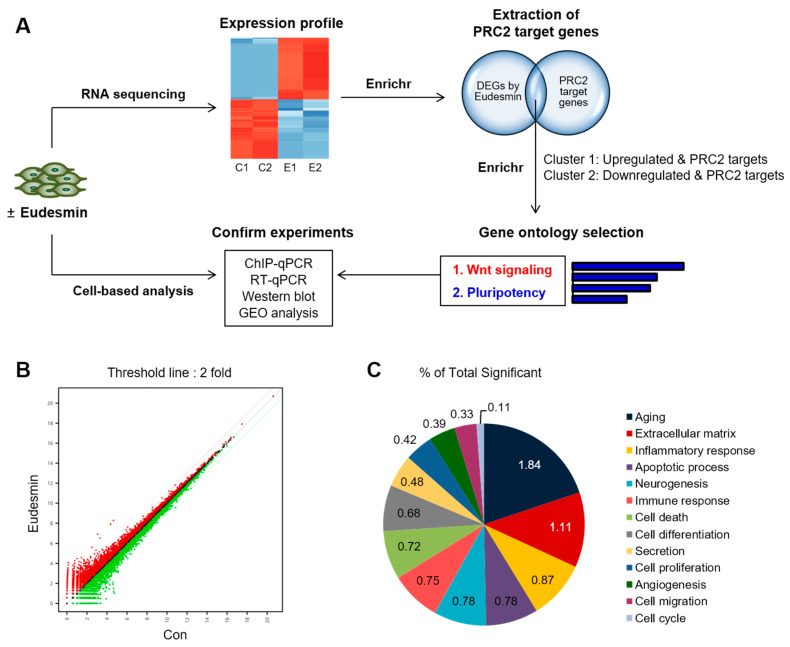
RNA sequencing analysis of eudesmin-treated 10T1/2 mesenchymal stem cells (**A**) Analysis flow to identify epigenetic effects of eudesmin. (**B**) Scatter plot of showing differentially expressed genes between control group and eudesmin-treated group (cut-off threshold: fold change > 2). (**C**) Gene category plot showing significantly changed genes related to each ontology (cut-off threshold: fold change > 2, *p*-value < 0.01).

**Figure 2 molecules-26-05665-f002:**
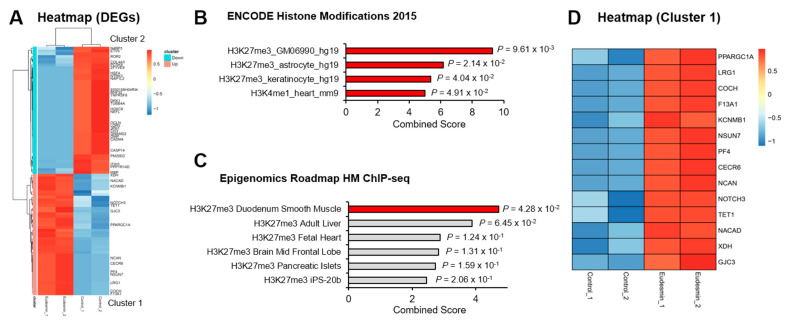
Enrichment of PRC2 target genes in eudesmin-mediated upregulated genes. (**A**) Heatmap of differentially expressed genes by eudesmin treatment (upregulated: 44 genes, downregulated: 46 genes). (**B**) Analysis of upregulated genes by eudesmin treatment with ENCODE Histone Modifications 2015 database in Enrichr. (**C**) Analysis of upregulated genes by eudesmin treatment with Epigenomics Roadmap HM ChIP-seq database in Enrichr. (**D**) Heatmap of PRC2 target genes upregulated by eudesmin (Cluster 1).

**Figure 3 molecules-26-05665-f003:**
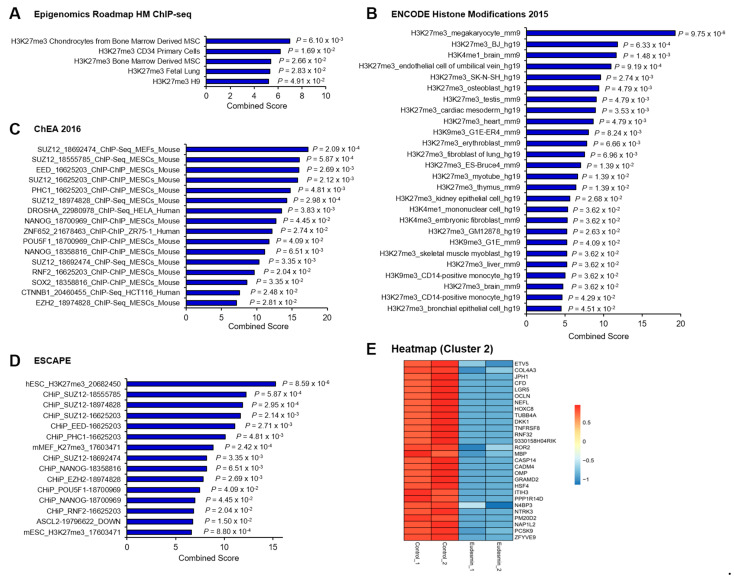
Enrichment of PRC2 target genes in differentially expressed genes by eudesmin treatment. (**A**) Analysis of downregulated genes by eudesmin treatment with Epigenomics Roadmap HM ChIP-seq database in Enrichr. (**B**) Analysis of downregulated genes by eudesmin treatment with ENCODE Histone Modifications 2015 database in Enrichr. (**C**) Analysis of downregulated genes by eudesmin treatment with ChEA 2016 database in Enrichr. (**D**) Analysis of downregulated genes by eudesmin treatment with ESCAPE database in Enrichr. (**E**) Heatmap of PRC2 target genes downregulated by eudesmin (Cluster 2).

**Figure 4 molecules-26-05665-f004:**
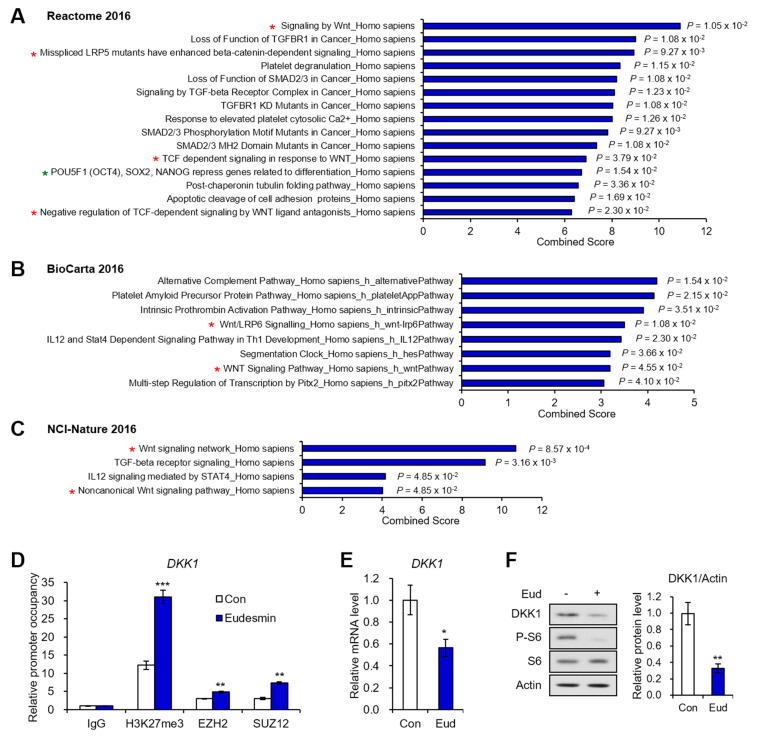
PRC2 target genes downregulated by eudesmin are involved in Wnt signaling. (**A**) Analysis of PRC2 target genes which are downregulated by eudesmin treatment with Reactome 2016 database in Enrichr. (**B**) Analysis of PRC2 target genes which are downregulated by eudesmin treatment with BioCarta 2016 database in Enrichr. (**C**) Analysis of PRC2 target genes which are downregulated by eudesmin treatment with NCI-Nature 2016 database in Enrichr. (**D**) H7 embryonic stem cells were treated with or without eudesmin (80 μM) for 24 h. ChIP assays were performed with IgG, H3K27me3, EZH2, and SUZ12 antibodies followed by real time PCR with primers for promoter region of *DKK1* genes. (**E**) The mRNA levels of *DKK1* gene in H7 embryonic stem cells treated with or without eudesmin (80 μM) for 24 h. (**F**) Immunoblot analysis of H7 embryonic stem cells treated with or without eudesmin (80 μM) for 24 h. Data represent means ± SEM for n = 3. * *p* < 0.05, ** *p* < 0.01, *** *p* < 0.001.

**Figure 5 molecules-26-05665-f005:**
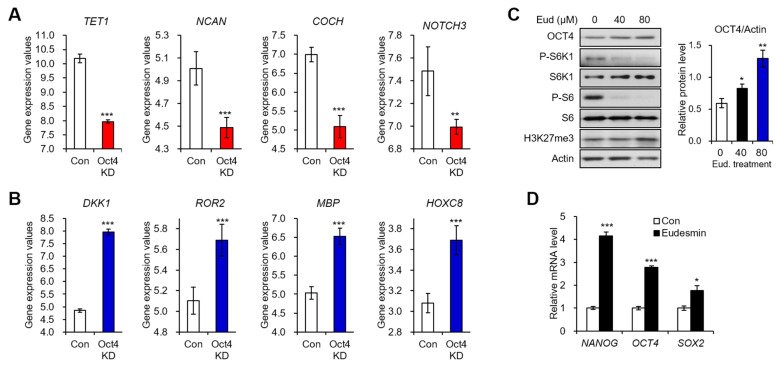
Eudesmin treatment promotes the expression of pluripotency marker genes. (**A**) Gene expression values of representative cluster 1 genes (*TET1*, *NCAN*, *COCH*, and *NOTCH3*) in Oct4-depleted embryonic stem cells (GDS1824). (**B**) Gene expression values of representative cluster 2 genes (*DKK1*, *ROR2*, *MBP*, and *HOXC8*) in Oct4-depleted embryonic stem cells (GDS1824). (**C**) Immunoblot analysis of H7 embryonic stem cells treated with or without eudesmin (40, 80 μM) for 24 h. (**D**) The mRNA levels of *NANOG*, *OCT4*, and *SOX2* genes in H7 embryonic stem cells treated with or without eudesmin (80 μM) for 24 h. Data represent means ± SEM for n = 3. * *p* < 0.05, ** *p* < 0.01, *** *p* < 0.001.

**Table 1 molecules-26-05665-t001:** PRC2 target genes upregulated by eudesmin (Cluster 1).

No.	Gene Symbol	Description	Eudesmin/Con	
			Fold Change	*p*-Value
1	*PPARGC1A*	Peroxisome proliferative activated receptor, gamma, coactivator 1 alpha	9.221	0.009
2	*LRG1*	Leucine-rich alpha-2-glycoprotein 1	4.717	0.002
3	*NPAS1*	Neuronal PAS domain protein 1	4.251	0.010
4	*COCH*	Cochlin	3.807	0.002
5	*F13A1*	Coagulation factor XIII, A1 subunit	3.807	0.002
6	*KCNMB1*	Potassium large conductance calcium-activated channel, subfamily M, beta member 1	3.018	0.005
7	*NSUN7*	NOL1/NOP2/Sun domain family, member 7	2.884	0.002
8	*PF4*	Platelet factor 4	2.884	0.002
9	*CECR6*	Cat eye syndrome chromosome region, candidate 6	2.884	0.002
10	*NCAN*	Neurocan	2.884	0.002
11	*NOTCH3*	Notch3	2.695	0.007
12	*TET1*	Tet methylcytosine dioxygenase 1	2.691	0.008
13	*NACAD*	NAC alpha domain containing	2.595	0.003
14	*GLIS1*	GLIS family zinc finger 1	2.482	0.010
15	*XDH*	Xanthine dehydrogenase	2.206	0.005
16	*GJC3*	Gap junction protein, gamma 3	2.198	0.009

**Table 2 molecules-26-05665-t002:** PRC2 target genes downregulated by eudesmin (Cluster 2).

No.	Gene Symbol	Description	Eudesmin/Con	
			Fold Change	*p*-Value
1	*ZFYVE9*	Zinc finger, FYVE domain containing 9	0.098	0.005
2	*PCSK9*	Proprotein convertase subtilisin/kexin type 9	0.155	0.005
3	*NAP1L2*	Nucleosome assembly protein 1-like 2	0.163	0.005
4	*PM20D2*	Peptidase M20 domain containing 2	0.178	0.001
5	*NTRK3*	Neurotrophic tyrosine kinase, receptor, type 3	0.195	0.005
6	*N4BP3*	NEDD4 binding protein 3	0.217	0.007
7	*PPP1R14D*	Protein phosphatase 1, regulatory (inhibitor) subunit 14D	0.218	0.005
8	*ITIH3*	Inter-alpha trypsin inhibitor, heavy chain 3	0.218	0.005
9	*HSF4*	Heat shock transcription factor 4	0.244	0.005
10	*GRAMD2*	GRAM domain containing 2	0.326	0.006
11	*OMP*	Olfactory marker protein	0.326	0.006
12	*CADM4*	Cell adhesion molecule 4	0.326	0.006
13	*CASP14*	Caspase 14	0.326	0.006
14	*MBP*	Myelin basic protein	0.419	0.008
15	*ROR2*	Receptor tyrosine kinase-like orphan receptor 2	0.425	0.006
16	*9330158H04RIK*	RIKEN cDNA 9330158H04 gene	0.491	0.006
17	*RNF32*	Ring finger protein 32	0.491	0.006
18	*TNFRSF8*	Tumor necrosis factor receptor superfamily, member 8	0.491	0.006
19	*DKK1*	Dickkopf WNT signaling pathway inhibitor 1	0.491	0.006
20	*TUBB4A*	Tubulin, beta 4A class IVA	0.491	0.006
21	*HOXC8*	Homeobox C8	0.491	0.006
22	*NEFL*	Neurofilament, light polypeptide	0.491	0.006
23	*OCLN*	Occludin	0.491	0.006
24	*LGR5*	Leucine rich repeat containing G protein coupled receptor 5	0.491	0.006
25	*CFD*	Complement factor D (adipsin)	0.491	0.006
26	*JPH1*	Junctophilin 1	0.491	0.006
27	*COL4A3*	Collagen, type IV, alpha 3	0.495	0.004
28	*ETV5*	Ets variant 5	0.499	0.007

## Data Availability

Data are contained within the article.

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
