# Peer review of "Transcriptomics-Based Repositioning of Natural Compound, Eudesmin, as a PRC2 Modulator"

_molecules, 2021, doi:10.3390/molecules26185665_

Round 1
Reviewer 1 Report
In this manuscript, the authors carry out RNA-seq analyses to find the epigenetic modulations upon eudesmin treatment. They found that PRC2 target genes downregulated by eudesmin are closely related to Wnt signaling and pluripotency. By ChIP-qPCR, they found that eudesmin treatment increased the occupancy of PRC2 components, EZH2 and SUZ12, and H3K27me3 level on the promoter region of Dkk1. In addition, they also showed that the expression of pluripotency markers, Oct4, Sox2, and Nanog, was upregulated upon eudesmin treatment. Thus, they conclude that pharmacological modulation of PRC2 dynamics by eudesmin might control Wnt signaling and maintain pluripotency of stem cells.
1) In Figure 1A, the authors show an analysis flow to identify epigenetic effects of eudesmin. It would be good to show the heatmap of DEGs with hierarchical clustering. In addition to highlight the PRC2 target genes in tables, authors could label the cluster I and cluster II in the heatmap of DEGs.
2) The authors show a gene category plot in Figure 1C suggesting global effects of eudesmin. Authors wrote that 80 uM of eudesmin was used to treat cells for RNA-seq analyses, why the authors choose this concentration? Could authors perform this experiment as a dose dependent manner?
3) According to their analyses, H3K27me3 occupies most of upregulated and downregulated genes (Figure 2A-2D). However, by prediction, they show that components of PRC2, such as SUZ12, EED, and EZH2, can target downregulated genes by eudesmin (Figure 2E and 2F). What about the upregulated genes?
4) It would be good to add p-values to the bar graphs in Figure 2.
5) The authors analyzed the occupancy of H3K27me3, EZH2 and SUZ12 on the promoter region of Dkk1 by ChIP-qPCR. Could the authors also check other target genes? Could the authors present the ChIP efficiency by percentage of input instead of fold change?
6) The authors show increased levels of OCT4, NANOG and SOX2 upon eudesmin treatment by western blot and RT-qPCR (Figure 4C and 4D). Why don’t those genes appear in upregulated genes with RNA-seq analyses?
7) The Figure 4 is bit complicated. With these results cannot conclude that eudesmin contributes to maintaining pluripotency. The y-axel in Figure 4A and 4B is “relative expression”. What is it relative to?
Author Response
In this manuscript, the authors carry out RNA-seq analyses to find the epigenetic modulations upon eudesmin treatment. They found that PRC2 target genes downregulated by eudesmin are closely related to Wnt signaling and pluripotency. By ChIP-qPCR, they found that eudesmin treatment increased the occupancy of PRC2 components, EZH2 and SUZ12, and H3K27me3 level on the promoter region of Dkk1. In addition, they also showed that the expression of pluripotency markers, Oct4, Sox2, and Nanog, was upregulated upon eudesmin treatment. Thus, they conclude that pharmacological modulation of PRC2 dynamics by eudesmin might control Wnt signaling and maintain pluripotency of stem cells.
We thank the reviewer for recognizing the potential importance of our findings. We feel our responses to his/her criticisms have improved the manuscript with several corrections. Our responses to the reviewer’s queries are described point-by-point below.
1) In Figure 1A, the authors show an analysis flow to identify epigenetic effects of eudesmin. It would be good to show the heatmap of DEGs with hierarchical clustering. In addition to highlight the PRC2 target genes in tables, authors could label the cluster I and cluster II in the heatmap of DEGs.
As the reviewer recommended, we added the heatmap of total DEG (current Figure 2A), Cluster 1 (current Figure 2D), and Cluster 2 (current Figure 3E). Accordingly, enrichr analysis data of Cluster 1 and Cluster 2 were separated as Figure 2 and 3.
2) The authors show a gene category plot in Figure 1C suggesting global effects of eudesmin. Authors wrote that 80 uM of eudesmin was used to treat cells for RNA-seq analyses, why the authors choose this concentration? Could authors perform this experiment as a dose dependent manner?
In our previous study (BBRC, 2018), we treated 10T1/2 cells with diverse concentration of eudesmin and found that 80 uM of eudesmin completely inhibited the action of S6K1. Thus, we used 80 uM of eudesmin for RNA-seq analysis.
3) According to their analyses, H3K27me3 occupies most of upregulated and downregulated genes (Figure 2A-2D). However, by prediction, they show that components of PRC2, such as SUZ12, EED, and EZH2, can target downregulated genes by eudesmin (Figure 2E and 2F). What about the upregulated genes?
We also compared the transcriptional components related with the upregulated genes upon eudesmin treatment using Enrichr database. However, we could not find any PRC2 components which are significantly related with upregulated genes. Thus, we focused on downregulated genes by eudesmin to assess the effects of eudesmin on PRC2-mediated gene regulation.
4) It would be good to add p-values to the bar graphs in Figure 2.
As the reviewer suggested, we added P-values to the bar graphs of Enrichr analysis in current Figure 2, 3, and 4.
5) The authors analyzed the occupancy of H3K27me3, EZH2 and SUZ12 on the promoter region of Dkk1 by ChIP-qPCR. Could the authors also check other target genes? Could the authors present the ChIP efficiency by percentage of input instead of fold change?
There were other target genes in Wnt-related PRC2 target genes downregulated by eudesmin like ROR2 or LGR5. However, DKK1 was the most frequently found gene in comprehensive analysis with Reactom2016 (current Figure 4A). Furthermore, analysis with BioCarta2016 showed that only DKK1 was shown in Wnt signaling-related categories (‘Wnt/LRP6 Signalling_Homo sapiens_h_wnt-lrp6Pathway’ and ‘WNT Signaling Pathway_Homo sapiens_h_wntPathway’). Thus, we focused on PRC2-mediated Dkk1 suppression upon eudesmin treatment.
Regarding the presentation of ChIP data, the occupancies of each factor were normalized with input and then the percentages of each factor against input were calculated relatively to the values of IgG-Con (presented as 1.00). We added this detailed calculation methods to ‘Materials and Methods’ section.
6) The authors show increased levels of OCT4, NANOG and SOX2 upon eudesmin treatment by western blot and RT-qPCR (Figure 4C and 4D). Why don’t those genes appear in upregulated genes with RNA-seq analyses?
RNA seq was performed with 10T1/2 mesenchymal stem cells, which are committed from pluripotent state into multipotent state. As OCT4, NANOG, and SOX2 are critical factors for pluripotent stem cells, their expression is extremely low in 10T1/2 cells. Hence, the fold changes of the genes are not accurate or undetectable in 10T1/2. For western blot and RT-qPCR, we used H7 embryonic stem cells, which are pluripotent and thereby highly express the three pluripotency marker genes.
7) The Figure 4 is bit complicated. With these results cannot conclude that eudesmin contributes to maintaining pluripotency. The y-axel in Figure 4A and 4B is “relative expression”. What is it relative to?
We used “values” that are presented in GEO profiles as the y axis of Figure 4A and 4B (Figure 5A and 5B in revised version). Samples within a GEO Datasets (GDS) refer to the same Platform, that is, a common set of elements are assayed. Calculations are computed on the ‘value’ column extracted from original Sample data tables. These value measurements are calculated in an equivalent manner for each Sample within a GDS. To make it clear, we corrected the label of y axis in Figure 4A and 4B (current Figure 5A and 5B) and their legends to “Gene expression values”.

Reviewer 2 Report
This study aims to investigate the epigenetic modulations of eudesmin in genome-wide level on mesenchymal stem cell line and elucidated its underlying mechanisms. This manuscript suggested that eudesmin treatment regulated PRC2 components, which promoted differentiate property of mesenchymal stem cell line, and decrease the expression of DKK1. In addition, eudesmin maintained embryonic stem cell through upregulated the expression of pluripotency markers, Oct4, Sox2, and Nanog. These findings described here is interesting. However, there are some research results need further explain in this manuscript.
- For Immunoblot analysis, please provide the quantitative calculations and internal control (such as actin or GAPDH).
- Did eudesmin affect other Wnt signaling pathway-related proteins?
- I suggest that the author should add H3K27me3 to further confirm the relationship between eudesmin and DKK1.
- In manuscript, the author mentioned that eudesmin increased the occupancy of EZH2 on DKK1 promoter region. However, earlier study demonstrated the inhibitory effect of eudesmin on EZH2. It’s confused.
- In "Discussion" section, the description of discussion should be improved; more details should be added in order to improve the understanding of present work.
Author Response
This study aims to investigate the epigenetic modulations of eudesmin in genome-wide level on mesenchymal stem cell line and elucidated its underlying mechanisms. This manuscript suggested that eudesmin treatment regulated PRC2 components, which promoted differentiate property of mesenchymal stem cell line, and decrease the expression of DKK1. In addition, eudesmin maintained embryonic stem cell through upregulated the expression of pluripotency markers, Oct4, Sox2, and Nanog. These findings described here is interesting. However, there are some research results need further explain in this manuscript.
We thank the reviewer for recognizing the potential importance of our findings. We feel our responses to his/her criticisms have improved the manuscript with several corrections. Our responses to the reviewer’s queries are described point-by-point below.
For Immunoblot analysis, please provide the quantitative calculations and internal control (such as actin or GAPDH).
We appreciate the reviewer for suggesting a good way to enhance the quality of our immunoblot data. We newly detected actin with the samples of current Figure 5C for the quantification and added the quantitative graphs of DKK1 relative to actin (current Figure 4F) and OCT4 relative to actin (current Figure 5C).
Did eudesmin affect other Wnt signaling pathway-related proteins?
There were other target genes in Wnt-related PRC2 target genes downregulated by eudesmin like ROR2 or LGR5. However, DKK1 was the most frequently found gene in comprehensive analysis with Reactom2016 (current Figure 4A). Furthermore, analysis with BioCarta2016 showed that only DKK1 was shown in Wnt signaling-related categories (‘Wnt/LRP6 Signalling_Homo sapiens_h_wnt-lrp6Pathway’ and ‘WNT Signaling Pathway_Homo sapiens_h_wntPathway’). Thus, we focused on PRC2-mediated Dkk1 suppression upon eudesmin treatment.
I suggest that the author should add H3K27me3 to further confirm the relationship between eudesmin and DKK1.
We conducted ChIP-qPCR to assess H3K27me3 level on DKK1-encoded region in DNA upon eudesmin treatment (current Figure 4D) as well as western blot to check the global level of H3K27me3 by eudesmin (current Figure 5C).
In manuscript, the author mentioned that eudesmin increased the occupancy of EZH2 on DKK1 promoter region. However, earlier study demonstrated the inhibitory effect of eudesmin on EZH2. It’s confused.
As the reviewer mentioned, the effects of eudesmin on EZH2 activity seems to be different for each cell and each gene. Hence, we demonstrated that eudesmin is not a simple inhibitor but precise modulator of EZH2-dependent gene regulation in discussion part. We added more detailed description about it at discussion section (page 8, lines 187-191).
In "Discussion" section, the description of discussion should be improved; more details should be added in order to improve the understanding of present work.
We agree that improvement of demonstration in discussion section can help better understanding of our study. Especially, the second paragraph seemed to be confusing thereby need accurate description about our present data comparing the previous study. We added more detailed demonstration on the effects of eudesmin on S6K1 and EZH2.
Round 2
Reviewer 1 Report
The revised manuscript has good flow, clear figures and corresponding figure legends. Two minor suggestions,
- The “cluster I and II” could be defined in the Fig. 2A.
- In the bar plot of Fig.5C, it is better to label the x-axis with “Eud. treatment”.
Author Response
The revised manuscript has good flow, clear figures and corresponding figure legends. Two minor suggestions,
- The “cluster I and II” could be defined in the Fig. 2A.
→ We added the names of Cluster 1 and 2 genes and defined them as Cluster 1 and 2 in Figure 2A, as suggested by the reviewer.
- In the bar plot of Fig.5C, it is better to label the x-axis with “Eud. treatment”.
→ We thank the reviewer for good suggestion. We added "Eud. treatment" to the X-axis label in Fig. 5C.
Reviewer 2 Report
The Authors have taken into considerations the Referee' suggestion. The manuscript can be accepted.
Author Response
We thank the reviewer for the constructive review of our manuscript.